# Increasing early phase clinical trials capacity in India

Jerin Jose Cherian, Aruvi Poomali, Aparna Mukherjee, Taruna Madan Gupta, Bikash Medhi, Shoibal Mukherjee, Alangudi Sankaranarayanan, Nilanjan Saha, Vikram Gota, Ramachandra Subbaraya Gudde, Monika Pahuja, Saibal Das, Nabendu Sekhar Chatterjee, Rubina Bose, Nilima Kshirsagar & Rajeev Singh Raghuvanshi
The Indian Council for Medical Research has established the ICMR-Network for Phase 1 Clinical Trials. Here we describe identification of the need to establish the Network and further elaborate on its vision, governance, and operational aspects.

## Background

India, with a population of 1·4 billion and its diverse disease burden, presents a unique opportunity for clinical research. While India accounts for nearly 20% of the global population, only 1·5% of global clinical trials are conducted in the country[1]. Between 2007–2018, fewer regulatory clinical trials were registered on the Clinical Trials Registry of India (CTRI) as compared to academic non–regulatory trials, with 21.4% of all trials being primarily industry-sponsored[2]. In contrast, 41% of clinical trials registered in ClinicalTrials.gov as of mid-2011 were industry-sponsored[3]. It is also worth noting that India has specifically shied away from early phase clinical trials. Indian innovators often pursue their early clinical development overseas, and international pharmaceutical giants are reluctant to engage with collaborators from India for early phase clinical development. According to a report, phase 1 clinical trial activity in India is limited owing to low technological innovation and low clinical research capacity[4].

Early–phase clinical trials are essential for evaluating new drugs' safety, tolerability, and pharmacokinetics. They provide valuable insights into the safety of potential therapeutic agents before they are advanced to larger–scale studies. Setting up capacity for Phase 1 clinical trials will allow clinical development of molecules that will meet the unmet medical needs of the population of India, especially when such molecules hold limited appeal for the pharmaceutical industry. It will also generate data about pharmacogenetics, and any impact of food–health systems on drug response[5]. In this comment we discuss the identification of the need to build early-phase clinical trial capacity in the country and the establishment of the Indian Council of Medical Research (ICMR)- Phase 1 Clinical Trial Network. We expect the network to not just develop leads of national health priority but also be of value for other countries seeking to build phase 1 clinical trial capacity.

## Trial trends from registries

Between 1999–2022, the World Health Organization—International Clinical Trials Registry Platform (ICTRP) registered 51,755 Phase 1 clinical

trials. Of these, 22,543 Phase 1 trials were conducted in the United States, while 2260 trials were registered from India (Fig. 1)[6]. Between 2008–2022 only 220 first–in–human phase 1 clinical trials were registered and conducted in India[7]. Various authors have also discussed that there may be misclassifications and inaccuracies in such registries, for example a few hundred studies were identified to be misclassified as Phase 1 clinical trials in the CTRI[7,8].

The number of Phase 1 clinical trials reported from China during a similar period from 2011–2020 is 2842. Chen et al. credit China's new and improved clinical trial system and drug innovation policy for this development[9]. The major new drug innovation programme for the 11th, 12th, and 13th Five-Year Plans have supported the development of innovative drugs and medical devices and helped to build the infrastructure of clinical trial platform compatible with international regulatory systems. Other reforms included the National Medical Products Administration (NMPA) emphasis on examination and approval based on clinical value. The reforms also included a 60 day acquiescence system for approval, relaxation of restrictions on imported drug approvals, and acceptance of clinical trial data from studies conducted abroad. Policy change has also encouraged development of biomedicine within the country[9].

Despite India's potential, several factors have hindered the conduct of early–phase trials, including regulatory challenges, infrastructure limitations, and a lack of skilled personnel[10]. A brief decline in clinical research activity in India after 2013 has been attributed to the regulatory reforms, as affirmed in a survey which explored the opinions and perceptions of clinical trial investigators regarding the new guidelines. Restrictions on the number of trials per investigator and AV consenting were the main issues[11–13]. The inadequate ecosystem for indigenous clinical trials was excusable in an era when biopharmaceutical innovation in India was uncommon, the regulatory processes were maturing, and most players focused on manufacturing generic drugs. However, in recent years this trend has seen a slight but exciting shift[2,7].

## Recent changes in the biopharmaceutical industry in India

Various factors have increased biopharmaceutical innovation in India. The trend began in the last decade with an increased willingness for risk from in-house biopharmaceutical innovators, and the lower profit margins on generic medicines as cheaper active pharmaceutical ingredients began to be imported from countries such as China[14]. The outsourcing of biopharmaceutical industry requirements to India for upstream drug discovery work has also bolstered the necessary expertise amongst Indian scientists[15]. Moreover, in the aftermath of the decline of clinical trials after 2013, the regulator has become proactive in clarifying concerns regarding the conduct

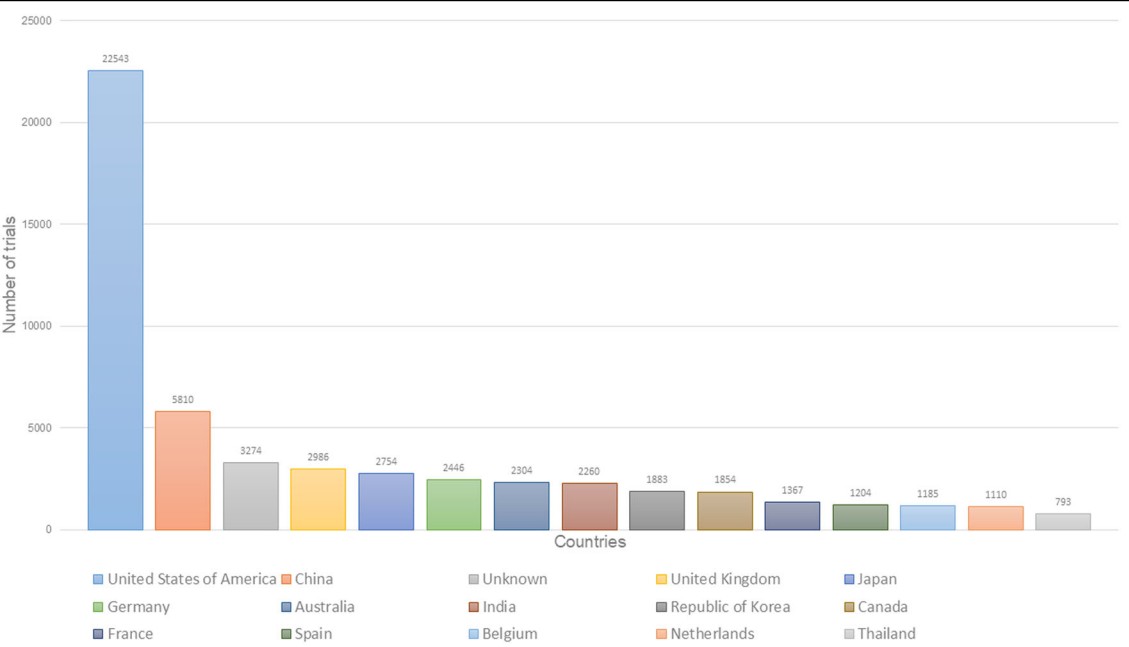

**Fig. 1 | Country–wise Phase 1 clinical trials registered at ICTRP (1999–2022).** According to ICTRP 22,543 Phase 1 trials have been conducted in the United States between 1999–2022, while only 2260 trial were registered from India.

of clinical trials in India[12]. Today, there are numerous success stories of biopharmaceutical innovation from India, targeting diseases that are of priority to the nation as well as the rest of the world[16].

Over one hundred companies are currently developing and marketing biosimilars across the country. Many of these companies are using their experience in this field to address endemic diseases, and this has resulted in indigenously produced monoclonal antibodies entering the market[17]. The development of India's first indigenous vaccine against SARS–CoV–2 is a public–private partnership success story[18]. More recently, collaborations between research institutes and domestic pharmaceutical industries are exploring innovative drug development models and enabling the market authorisation of new molecules[16]. An example of this model is a version of CAR T–cell therapy available at a cost one–tenth of comparable commercial products[19].

### Indian pharmaceutical ecosystem in transition
India must utilize the opportunity presented by the recent developments discussed above for multiple reasons. The first and foremost reason is economic – a strong industry can generate jobs, advance trade, and improve the GDP. Another is that the clinical trial ecosystem in India should be capable of undertaking clinical development of leads that are of relevance to India. Third is the fact that while there is an increasing trend in the number of clinical trials happening globally, there is an under–representation of diverse ethnic groups in these trials[5,20]. Lastly, it is well documented that many multinational enterprises that operate within knowledge–intensive industries expand their influence in emerging economies to strengthen their competitive advantage. Munjal et al. introduce the concept of reverse asset augmentation to highlight the behaviour whereby a multinational enterprise identifies a strategic asset from which knowledge transfer is possible. Knowledge transfer from these newly acquired subsidiaries, termed reverse knowledge transfer, occurs through complex pathways and helps maximise the gains for the multinational enterprise[21]. Recently the Indian Council of Medical Research (ICMR), at the behest of the Government of India, has

begun to establish world–class public–sector infrastructure for early phase clinical development of health products.

### Vision
The ICMR Network of Phase 1 Clinical Trials is designed to create a robust infrastructure that supports the rigorous demands of early phase clinical trials and aims to foster a conducive environment for these trials in India, thereby contributing to global drug discovery and development. Utilization of this capacity built by ICMR is expected to derisk the failure inherent in early phase product development.

Highlights of the ICMR Network of Phase 1 Clinical Trials are listed in Table 1.

### Leadership and governance
An Inter–Ministerial Steering Committee (IMSC) of Secretaries from Department of Biotechnology (DBT), Department of Science and Technology (DST), Defence Research and Development Organization (DRDO), Department of Atomic Energy (DAE), Department of Pharmaceuticals (DoP), and Department of Scientific and Industrial Research (DSIR) oversees the high–level strategy and monitoring mechanism. The Centre for Advanced Research Advisory Committee (CAC) oversees the performance and operations of partnering institutions, while the Health Product Screening Committee (HPSC) oversees the identification of leads of national health priorities that are ready for Phase 1 clinical trials. The Central Drugs Standard Control Organization (CDSCO) has identified the initiative as a step towards encouraging clinical research and innovation in the country and has assured timely support for regulatory guidance as per rules.

### Operational aspects
Currently, four partnering institutions have been identified through a rigorous vetting process. Institutions from across the nation were invited to apply to be a part of the initiative through an Expression of Interest (Fig. 2).

**Table 1 | Highlights of the ICMR Network of Phase 1 clinical trials**

| | Highlights of the ICMR Network of Phase 1 Clinical Trials | |
|---|---|---|
| 1. | Infrastructure Development | The establishment of dedicated early–phase trial centres equipped with state–of–the–art facilities is a cornerstone of this initiative. These centres will provide the necessary environment for conducting complex trials, including advanced imaging, laboratory services, pharmacokinetics, and data management systems. |
| 2. | Manpower Training | Recognizing that skilled personnel are essential for the success of early–phase trials, ICMR is investing in extensive training programmes. These programmes will cover a wide range of topics, from clinical trial design, ethical considerations, national and international regulatory requirements, and data analysis. |
| 3. | Standardization and Protocol Development | To ensure consistency and high–quality outcomes, ICMR is supporting protocol development and adopting standard procedures, SOPs, and best practices for early–phase trials. This will facilitate smoother operations and help in maintaining global standards. |
| 4. | Collaborations and Partnerships | The initiative also emphasizes the importance of collaborations with research institutions and pharmaceutical companies, agnostic of their origin. By fostering these partnerships with innovators, ICMR aims to enhance knowledge exchange and integrate global advancements into India's research framework. |
| 5. | Support for innovators | The initiative is expected to extend regulatory and operational support to innovators, especially to academia and start-ups with limited experience in early phase drug development. The network is also expected to be flexible enough to help these innovators and partners meet the requirements of regulatory agencies across the world. |
| 6. | Cost and risk sharing | Clinical development is the most cost intensive part of pharmaceutical development. With ICMR sharing the costs and risks of clinical trials, the burden borne by the innovators will be reduced. |

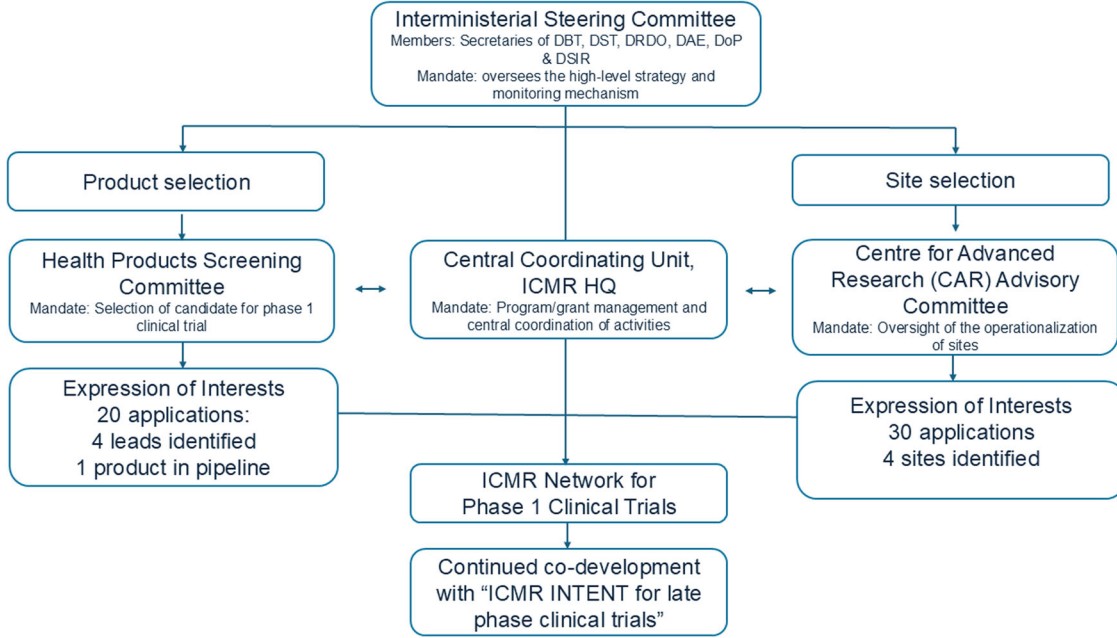

**Fig. 2 | Organogram of the ICMR Network of Phase 1 Clinical Trials.** The Interministerial Steering Committee oversees the high-level strategy and monitoring mechanism of the ICMR Network for Phase 1 Clinical Trials. The sites and leads were identified from two separate Expressions of Interests that were floated. The Health Products Screening Committee is responsible for the selection of the leads for the Phase 1 Clinical Trials and the CAR Advisory Committee oversees the operationalization of the sites. The Central Coordination Unit at ICMR Headquarters is responsible for the programme management and coordination. Continued co-development of the leads will be through the ICMR INTENT network. Abbreviations: DBT Department of Biotechnology, DST Department of Science and Technology, DRDO Defence Research and Development Organization, DAE Department of Atomic Energy, DoP Department of Pharmaceuticals, DSIR Department of Scientific and Industrial Research, ICMR HQ Indian Council of Medical Research Headquarters, CAR Centre for Advanced Research, INTENT Indian Clinical Trial and Education Network.

A shortlist was prepared from 30 eligible applications based on a scoring system which assessed the competency of the investigators, infrastructure available at the site, the availability of a trained and multidisciplinary team, their previous experience with early phase clinical trials, and the institutional support availability. The CAC reviewed these scores and identified the top ten institutes following an independent review. These institutions were then invited for a discussion, following site visits. A strengths, weaknesses, opportunities, and threats (SWOT) analysis of these proposed facilities was then conducted, following which the four sites were identified.

The four institutions are: Advanced Centre for Treatment, Research and Education in Cancer, Navi Mumbai; King Edward Memorial Hospital & Seth Gordhandas Sunderdas Medical College, Mumbai; Postgraduate

Institute of Medical Education and Research, Chandigarh; and Sri Ramaswamy Memorial Medical College Hospital and Research Centre, Kattankulathur. They have a combined experience of conducting 25 Phase 1 clinical trials. These four institutions signed an agreement with the Central Coordinating Unit at ICMR on 1st May 2024 confirming their participation in the ICMR Network of Phase 1 Clinical Trials.

ICMR expects to bolster the existing systems at these institutions through a research grant spread over a period of 5 years. The grant will be used to establish infrastructural and manpower capacity. Besides funds for capacity building, ICMR will also support trials identified by the HPSC with additional resources for project specific funds.

The roles of the sites have been determined based on the SWOT analysis and recommendations from the CAC. By working as a Network of sites for Phase 1 clinical trials, they will be able to build on each other's strengths, and through resource and knowledge–sharing, the sites will learn from the best practices of each other. The infrastructure development phase has been completed, SOPs have been developed to harmonize practices, and a road map for workshops and periodic in–house training for all sites has been set up. The readiness of ICMR with agreements and flexibility in the norms is expected to encourage innovation by providing technical and funding support to young innovators with little experience to take their lead to the clinical development phase. The ICMR Central Coordinating Unit's regular site visits to monitor these studies will also ensure regulatory compliance for clinical practices and data quality. These facilities are expected to plan and conduct clinical trials across all product categories and therapeutic classes. The trial sites are positioned as national assets that are encouraged to be utilised by all partners, including well-established ones from the pharmaceutical industry.

## Finding investigational products for early clinical development

Work is currently progressing at the trial sites on oncology, vaccine, and cell & gene therapy leads. The leads were identified by floating another Expression of Interest to identify the lead molecules to be taken up for Phase 1 clinical trials by the Network. Molecules were selected by the HPSC, from 20 applications, based on a set of criteria including the unmet need, their readiness for clinical development, existence of a competitive edge, and the identification of an eventual market (Fig. 2).

As these molecules enter Phase 1 clinical trials, the pipeline will be kept running in collaboration with research, academic and industry partners, through another Expression of Interest and the MedTech Mitra portal. The MedTech Mitra portal (https://medtechmitra.icmr.org.in/) is an effort to support innovators from conception to delivery of product development, including support for early phase clinical trials. ICMR's Indian Clinical Trial and Education Network (INTENT) is expected to provide the platform for late phase clinical trials[22].

## The way forward

The initiative is not without its fair share of challenges and limitations. A fragmented research environment with poor communication amongst stakeholders is one of the primary challenges of public partnerships[23]. Efforts are in place to ensure coordination and cooperation among the four sites for various activities such as data management, regulatory dossier preparations, and protocol development are seamless. The network encourages transparency in all its proceedings to try and address such communication-related issues. The role of bureaucracy, leading to inefficient timelines of clinical trials as discussed extensively in literature, is a possible stumbling block[24–26]. In any newly established system, obstacles are

to be expected, but can be overcome. The first few trials under the initiative are scheduled to be launched in 2025. Future evaluation of the initiative through the assessment of key performance indicators will help in course correction and strengthening of the network.

**Reporting summary.** Further information on research design is available in the Nature Portfolio Reporting Summary linked to this article.

**Jerin Jose Cherian**[1,2] ✉, **Aruvi Poomali** [ORCID][1] ✉, **Aparna Mukherjee**[1] ✉, **Taruna Madan Gupta**[1] ✉, **Bikash Medhi**[3], **Shoibal Mukherjee**[4], **Alangudi Sankaranarayanan** [ORCID][5,14], **Nilanjan Saha**[6], **Vikram Gota**[7], **Ramachandra Subbaraya Gudde**[8,14], **Monika Pahuja**[9], **Saibal Das**[10], **Nabendu Sekhar Chatterjee**[11,14], **Rubina Bose**[13,14], **Nilima Kshirsagar**[12,14] **& Rajeev Singh Raghuvanshi**[13,14]

[1]Division of Development Research, Indian Council of Medical Research Headquarters, New Delhi, India. [2]Department of Global Public Health, Karolinska Institutet, Stockholm, Sweden. [3]PGIMER, Chandigarh, India. [4]Medanta, Gurugram, India. [5]Vivo Bio Tech Ltd, Hyderabad, Telangana, India. [6]Department of Translational & Clinical Research, Jamia Hamdard, New Delhi, India. [7]ACTREC, Navi Mumbai, India. [8]Indian Institute of Science, Bengaluru, India. [9]Division of Discovery Research, Indian Council of Medical Research Headquarters, New Delhi, India. [10]ICMR-Centre for Ageing & Mental Health, Kolkata, India. [11]National Institute for Research in Bacterial Infection, Kolkata, India. [12]Former National Chair Clinical Pharmacology, ICMR, New Delhi, India. [13]CDSCO, New Delhi, India. [14]These authors contributed equally: Alangudi Sankaranarayanan, Ramachandra Subbaraya Gudde, Nabendu Sekhar Chatterjee, Rubina Bose, Nilima Kshirsagar and Rajeev Singh Raghuvanshi.
✉e-mail: cherian.jj@icmr.gov.in; aruvipoomali@gmail.com; aparna.sinha.deb@icmr.gov.in; guptatnirrh@icmr.gov.in

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

## Funding

## Competing interests
The authors declare no competing interests.
