## [Peer Review file · Communications Medicine]

National Assets for Early Phase Clinical Trials Capacity in India

Corresponding Author: Dr Aruvi Poomali

Version 0:

Reviewer comments:

Reviewer #1

(Remarks to the Author)

The authors have done a nice job of summarizing a transition that is taking place in the Indian pharmaceutical landscape, and in ICMR's recent initiative of identifying hospitals that can conduct First-in-human Phase 1 clinical trials in India. The latter has been a weak spot in the country's trial ecosystem, and therefore is a valuable contribution to taking this ecosystem to another level.

Minor comments:

1. ... paper published in 2020 showed that between 2007–2018, fewer regulatory clinical trials were registered on the Clinical Trials Registry of India (CTRI) as compared to academic non-regulatory trials [How does this compare with ClinicalTrials.gov, for instance? A quick check may provide the answer, and enrich the sentence] .
2. The authors reported that of the regulatory trials, majority were sponsored by the pharmaceutical industry [That is not surprising, and may be a universal phenomenon. It may not be easy to determine this at other registries, but a comment could be made.]
3. The authors concluded that restrictions on the number of trials per investigator and AV consenting were the foremost points of contention [These restrictions were for a brief period only, and therefore should be commented upon].
4. The reluctant [a better word may be 'inadequate'] ecosystem for indigenous clinical trials was excusable in an era when pathbreaking Indian biopharmaceutical innovation was uncommon...
5. the regulatory checks were in their maturation phase [this phrase could be improved perhaps],
6. and most Indian players focused on manufacturing generic products of medicines. ('Generic drugs' may be a better phrase)
7. The offshore ('offshore' is redundant since 'outsourcing... to India' conveys the meaning) outsourcing of biopharmaceutical industry requirements to India for upstream drug discovery work has also bolstered the necessary expertise amongst Indian scientists.¹³
8. Moreover, in the aftermath of the decline of clinical trials after 2013, the regulator has become proactive in training and clearing ambiguity (in training and clearing ambiguity: unclear) regarding the conduct of clinical trials in India.¹
9. 'India in transition' would perhaps be better as 'Indian pharma ecosystem in transition'.
10. Reverse asset augmentation and reverse knowledge transfer: most readers would not understand these phrases. They should be explained/replaced.
11. Collaborations and Partnerships: The initiative also emphasizes the importance of collaborations of research institutions and pharmaceutical companies. Is the collaboration between research institutions and multinationals? If so, this could be clarified.
12. Figure 2: There are 2 boxes listing 'Cycle 1'. The box on the right hand side has been explained previously, but not the one on the left until later in the manuscript. It is confusing.

13. The first few trials under the initiative are scheduled to be launched later this year. Since it is already mid November and the article is unlikely to appear before 2025, it may be better to say 'later in 2024'.

Reviewer #2

(Remarks to the Author)

I have read the commentary titled "National Assets for Early Phase Clinical Trials Capacity in India." While the initiative to establish the ICMR Network of Phase 1 Clinical Trials is highly promising, there are a few aspects in the manuscript that require clarification:

1. "The number of Phase 1 clinical trials reported from China during a similar period from 2011–2020 is 2,842. Chen et al. credit China's new and improved clinical trial system and drug innovation policy for this development." – This sentence appears to give a very superficial overview – The authors may consider elaborating on some actionable pointers in the new and improved system that can be implemented by India.
2. The number of Phase 1 clinical trials conducted in countries such as Japan, UK, Germany and Australia is very similar to India. Are regulatory challenges, lack of infrastructure and lack of skilled personnel the only obstacles that need to be targeted. If so then why do these countries lag behind? We must also think of global level challenges that are not generic to developing countries. What will be done by ICMR to encounter them? What can India do differently than Japan, UK, Germany and Australia?
3. What will be done to ensure that the Phase I trials conducted in India meet the Global level regulatory requirements?
4. "India's decision to waive local clinical trials for drugs approved in countries like the U.S., U.K., Japan, and the European Union represents a pivotal regulatory shift. While the move aims to expedite access to essential medications and enhance market availability, it raises significant concerns about patient safety, research and development (R&D), and broader healthcare implications." – How will this impact the ICMR Network of Phase 1 clinical Trials?
5. What is the long-term global implication of this initiative?
6. What are some success stories from the 2,260 Phase 1 studies being conducted in India that motivate this Initiative by the ICMR?
7. Can Phase 1 clinical trials of Global Pharmaceutical MNC's be outsourced to India?
8. Majority of the support only appears to be offered to "Academia and start-ups with limited experience". What kind of support will be offered by the ICMR to bigger pharmaceutical companies?

Overall the manuscript needs to bring out a clarity on actionable insights, and what is ICMR doing to ensure that India can make an impact like China.

Version 1:

Reviewer comments:

Reviewer #1

(Remarks to the Author)

While this may be a universal phenomenon, it worth noting that India...
to change to

While this may be a universal phenomenon, it is worth noting that India...

Reviewer #2

(Remarks to the Author)

Thank you for the opportunity to re-review this manuscript.

The authors have submitted a rebuttal letter addressing the provided comments. However, for certain comments, the authors have only provided responses in the letter, without incorporating the corresponding content into the manuscript itself.

1. What will be done to ensure that the Phase I trials conducted in India meet the Global level regulatory requirements?
2. "India's decision to waive local clinical trials for drugs approved in countries like the U.S., U.K., Japan, and the European Union represents a pivotal regulatory shift. While the move aims to expedite access to essential medications and enhance market availability, it raises significant concerns about patient safety, research and development (R&D), and broader healthcare implications." – How will this impact the ICMR Network of Phase 1 clinical Trials?
3. What is the long-term global implication of this initiative?
4. Can Phase 1 clinical trials of Global Pharmaceutical MNC's be outsourced to India?
5. Majority of the support only appears to be offered to "Academia and start-ups with limited experience". What kind of support will be offered by the ICMR to bigger pharmaceutical companies?

The Authors have not provided a suitable response for the following comment

1. The number of Phase 1 clinical trials conducted in countries such as Japan, UK, Germany and Australia is very similar to India. Are regulatory challenges, lack of infrastructure and lack of skilled personnel the only obstacles that need to be targeted. If so then why do these countries lag behind? We must also think of global level challenges that are not generic to

developing countries. What will be done by ICMR to encounter them? What can India do differently than Japan, UK, Germany and Australia?

Version 2:

Reviewer comments:

Reviewer #2

(Remarks to the Author)

Thank you for providing me with an opportunity to re review the manuscript. Please find my comments below on the revisions.

1. The authors have declined to discuss regarding the global level challenges which developed countries who have the same number of Phase 1 Trials as India face – “stating that it is beyond the scope of the manuscript”. Such a rebuttal is not acceptable in an International Journal that caters to the Global Community of HCP’s

2. “The network is expected to cater to clinical trials for regulatory submission across the world. The regulatory requirement of the Central Licencing Authority of each country is different and the Network is expected to be flexible enough to meet these requirements.” – This is just mentioned in the rebuttal letter and not in the manuscript.

Response to referees:

Dear Reviewers,

We thank you for your time and effort in reviewing our paper. We appreciate your valuable comments and have incorporated the same to improve the article. Please find our point-by-point response below.

Tabular Summary of Response to Reviewer Comments

Reviewer #1 comments:	Response
The authors have done a nice job of summarizing a transition that is taking place in the Indian pharmaceutical landscape, and in ICMR's recent initiative of identifying hospitals that can conduct First-in-human Phase 1 clinical trials in India. The latter has been a weak spot in the country's trial ecosystem, and therefore is a valuable contribution to taking this ecosystem to another level.	We thank you for reviewing our article and taking the time to share your valuable insights and suggestions. We have made revisions as advised to improve the article.
1. ... paper published in 2020 showed that between 2007–2018, fewer regulatory clinical trials were registered on the Clinical Trials Registry of India (CTRI) as compared to academic non–regulatory trials [How does this compare with ClinicalTrials.gov, for instance? A quick check may provide the answer, and enrich the sentence] .	The following revision has been made. “A paper published in 2020 showed that between 2007–2018, fewer regulatory clinical trials were registered on the Clinical Trials Registry of India (CTRI) as compared to academic non–regulatory trials. The authors reported that 21.4% of all trials were primarily industry sponsored. A comparison with the sponsorship of clinical trials registered in ClinicalTrials.gov show that 41% of the trials as of mid-2011 were industry sponsored.”
2. The authors reported that of the regulatory trials, majority were sponsored by the pharmaceutical industry [That is not surprising, and may be a universal phenomenon. It may not be easy to determine this at other registries, but a comment could be made.]	We have made a note in the text (highlighted) that the sponsorship by industry may be a universal phenomenon.
3. The authors concluded that restrictions on the number of trials per investigator and AV consenting were the foremost points of contention [These restrictions were for a brief period only, and therefore should be commented upon.]	We have made a note in the text (highlighted) that the decline in clinical research due to these points were of a brief duration. The paragraph ends by noting an exciting shift and the article further goes on to elaborate on the shift.

4. The reluctant [a better word may be 'inadequate'] ecosystem for indigenous clinical trials was excusable in an era when pathbreaking Indian biopharmaceutical innovation was uncommon...	We have replaced the word.
5. the regulatory checks were in their maturation phase [this phrase could be improved perhaps],	We have improved the phrase and the sentence now reads as follows: "The inadequate ecosystem for indigenous clinical trials was excusable in an era when path breaking Indian biopharmaceutical innovation was uncommon, the regulatory processes were maturing, and most Indian players focused on manufacturing generic drugs."
6. and most Indian players focused on manufacturing generic products of medicines. ('Generic drugs' may be a better phrase)	We have replaced the phrase with "Generic drugs" as advised.
7. The offshore ('offshore' is redundant since 'outsourcing... to India' conveys the meaning) outsourcing of biopharmaceutical industry requirements to India for upstream drug discovery work has also bolstered the necessary expertise amongst Indian scientists.	We have improved the phrase by removing "offshore".
8. Moreover, in the aftermath of the decline of clinical trials after 2013, the regulator has become proactive in training and clearing ambiguity (in training and clearing ambiguity: unclear) regarding the conduct of clinical trials in India.	We have revised the sentence to give more clarity. It now reads as follows: "Moreover, in the aftermath of the decline of clinical trials after 2013, the regulator has become proactive in clarifying concerns regarding the conduct of clinical trials in India."
9. 'India in transition' would perhaps be better as 'Indian pharma ecosystem in transition'.	We have replaced the subheading as advised.
10. Reverse asset augmentation and reverse knowledge transfer: most readers would not understand these phrases. They should be explained/replaced.	We acknowledge your concern and have explained the phrases as given below: "Munjal et al. introduce the concept of reverse asset augmentation to highlight the behaviour whereby a multinational enterprise identifies a strategic asset from which knowledge transfer is possible. Knowledge transfer from these newly acquired subsidiaries, termed reverse knowledge transfer, occurs through complex

	pathways and helps maximise the gains for the multinational enterprise.”
11. Collaborations and Partnerships: The initiative also emphasizes the importance of collaborations of research institutions and pharmaceutical companies. Is the collaboration between research institutions and multinationals? If so, this could be clarified	We would like to clarify that the network is open to collaboration and the manuscript now reads as follows: “Collaborations and Partnerships: The initiative also emphasizes the importance of collaborations with research institutions and pharmaceutical companies, agnostic of their origin. By fostering these partnerships with innovators, the ICMR aims to enhance knowledge exchange and integrate global advancements into India's research framework.”
12. Figure 2: There are 2 boxes listing ‘Cycle 1’. The box on the right hand side has been explained previously, but not the one on the left until later in the manuscript. It is confusing.	References to the image have been included at both instances where the contents of the image are referenced. Additionally, we have improved the images to avoid confusion.
13. The first few trials under the initiative are scheduled to be launched later this year. Since it is already mid November and the article is unlikely to appear before 2025, it may be better to say ‘later in 2024’.	We have changed the phrase to reflect a revised timeline.

Reviewer #2 comments:	Response
1. “The number of Phase 1 clinical trials reported from China during a similar period from 2011–2020 is 2,842. Chen et al. credit China’s new and improved clinical trial system and drug innovation policy for this development.” – This sentence appears to give a very superficial overview – The authors may consider elaborating on some actionable pointers in the new and improved system that can be implemented by India.	We have addressed this comment by including the following explanation: “The Major New Drug Innovation Program for the Chinese 11th, 12th, and 13th Five-Year Plans have supported the development of innovative drugs and medical devices, and building of clinical trial platforms compatible with international regulatory systems. Additionally, a series of reforms including the National Medical Products Administration (NMPA) emphasis on examination and approval based on clinical value were introduced. The reforms also included a 60-day acquiescence system for approval, relaxation of restrictions on imported drug approvals, and acceptance of clinical trial data from studies conducted abroad. Policy change has also

	encouraged development of biomedicine in the country.”
2. The number of Phase 1 clinical trials conducted in countries such as Japan, UK, Germany and Australia is very similar to India. Are regulatory challenges, lack of infrastructure and lack of skilled personnel the only obstacles that need to be targeted. If so then why do these countries lag behind? We must also think of global level challenges that are not generic to developing countries. What will be done by ICMR to encounter them? What can India do differently than Japan, UK, Germany and Australia?	Please find our response below: While the number of Phase 1 clinical trials conducted in countries such as Japan, UK, Germany and Australia appear similar to India, we believe this could be due to a variety of reasons. A study has previously shown possible misclassifications and inaccuracies in one such registry and reports that only 220 First-in-human phase 1 clinical trials were conducted in India between 2008-2022. (reference #7) Notwithstanding these inaccuracies, the countries would still remain incomparable without an in-depth analysis the proportion of funding that goes into pharmaceutical R&D and the percentage of GDP that is earmarked for drug development. At the European Federation for Exploratory Medicines Development Lyon Conference 2019, experts identified the changing landscape of early clinical development and the challenges created by transformative technologies that are increasingly a part of trial designs. Newer agents such as biologicals also present challenges in early phase drug development. (https://www.frontiersin.org/journals/pharmacology/articles/10.3389/fphar.2019.01377/full) However, the CIOMS 2021 consensus on “Clinical research in resource-limited settings” identify a lack of conducive environment, challenges in implementing international standards and trust building as the major hurdles to good quality clinical research. (https://cioms.ch/publications/product/clinical-research-in-low-resource-settings/) Further, a work of Burt et al. (referenced in our article) has extensively researched the challenges to the clinical research environment in India. The domains they have identified include Regulatory (government, ethics committees, monitors, auditors), Professional (industry, academia, health care: clinicians, investigators, research staff), and Public (patient advocacy groups, NGOs, the media).(Reference #24) Thus we can

	see that the challenges faced by the high income countries in early phase clinical trials might be economic as well as scientific and we believe that the ICMR Network for Phase I clinical trials can address this through capacity building and support for the development of molecules of national health priority.
3. What will be done to ensure that the Phase I trials conducted in India meet the Global level regulatory requirements?	Please find our response below: The ICMR Central Coordinating Unit has so far conducted two major workshops for the ICMR Network for Phase 1 Clinical Trials. The first workshop was on quality systems and SOPs. The sites have since then harmonized the SOPs across all centres. The second workshop titled “The ICMR-PGIMER Workshop of Phase 1 Clinical Trials on Regulatory Aspects” was held with experts from two regulatory agencies (CDSCO and FDA) and industry leaders. The roadmap for future workshops and periodic in-house training for all sites has also been set up and has been mentioned in the text: “The infrastructure development phase has been completed, SOPs have been developed to harmonize practices, and a road map for workshops and periodic in-house training for all sites has been set up”.
4. “India's decision to waive local clinical trials for drugs approved in countries like the U.S., U.K., Japan, and the European Union represents a pivotal regulatory shift. While the move aims to expedite access to essential medications and enhance market availability, it raises significant concerns about patient safety, research and development (R&D), and broader healthcare implications.” – How will this impact the ICMR Network of Phase 1 clinical Trials?	Please find our response below: “The ICMR Network of Phase 1 Clinical Trials is designed to create a robust infrastructure that supports the rigorous demands of early phase clinical trials and aims to foster a conducive environment for these trials in India, and thereby contribute to global drug discovery and development. Utilization of this capacity built by ICMR is expected to derisk the failure inherent in early phase product development.” The Network is expected to aid in infrastructure development, manpower training, standardization of processes, collaborations and partnerships, support for innovators and cost and risk sharing. According to the authors, India’s decision to

	waive local clinical trials for drugs approved in countries for five categories of drugs (namely orphan drugs for rare diseases, gene and cellular therapies, pandemic related drugs, defense-specific treatments and drugs with significant therapeutic advancements) is not expected to impact the activities of Phase 1 clinical trials. While the decision will improve access to certain drugs under certain condition, the Network, moving on a different trajectory will help build indigenous capacity for drug development.
5. What is the long-term global implication of this initiative?	Please find our response below: Once established, the ICMR Network for Phase 1 clinical trials will not only build capacity for development of molecules of national priority but also be a beacon for drug development for LMICs, especially in the South-East Asian region. In their paper, Pant et al. elaborate on how the emerging Indian model - which includes the ICMR Network for Phase 1 Clinical Trials - will help LMICs in south east Asia breakaway from their dependency on high income countries for solutions to their national health problems.(Reference #16) In addition, the network is also expected to make India a go – to destination for early phase clinical trials from across the world.
6. What are some success stories from the 2,260 Phase 1 studies being conducted in India that motivate this Initiative by the ICMR?	Please find our response below: The success stories from these trials have been elaborated by Pant et al., in their paper, which we have referenced in the following section. The highlights include development of Centchroman, Risorine, ROTAVAC, COVAXIN, 2-DG, and NexCAR19. (Reference #16 in the manuscript) “Today there are numerous success stories of biopharmaceutical innovation from India, targeting diseases that are of priority to the nation as well as the rest of the world. (Reference #14 in the manuscript)More than a hundred companies are currently developing and

	marketing biosimilars across the country. Many of these companies are using their experience in this field to address endemic diseases and this has resulted in indigenously produced monoclonal antibodies entering the market. (Reference #15 in the manuscript) The development of the India's first indigenous vaccine against SARS-CoV-2 is a public-private partnership success story. (Reference #16 in the manuscript) More recently, collaborations of research institutes with domestic pharmaceutical industries are exploring innovative drug development models and enabling the market authorisation of new molecules. (Reference #14 in the manuscript) An example of this model is the home-grown version of CAR T-cell therapy available at a cost one-tenth of comparable commercial products."
7. Can Phase 1 clinical trials of Global Pharmaceutical MNC's be outsourced to India?	Please find our response below: The ICMR Network for Phase 1 Clinical Trials is expected to make India a go – to destination for early phase clinical trials from across the world. While the centres of the Network are expected to prioritize the molecules of national health priority identified and funded by ICMR, the centres are open to all early phase drug development agnostic of their origin.
8. Majority of the support only appears to be offered to "Academia and start-ups with limited experience". What kind of support will be offered by the ICMR to bigger pharmaceutical companies?	Please find our response below: While the ICMR Network for Phase 1 Clinical Trials is expected to support innovators with limited experience, the centres are open to all early phase drug development agnostic of their origin. The manuscript has been edited to reflect this. "4. Collaborations and Partnerships: The initiative also emphasizes the importance of collaborations with research institutions and pharmaceutical companies, agnostic of their origin. By fostering these partnerships with

	innovators, the ICMR aims to enhance knowledge exchange and integrate global advancements into India's research framework. 5.Support for innovators: The initiative is expected to extend regulatory and operational support to innovators, especially to academia and start-ups with limited experience in early phase drug development.”
Overall the manuscript needs to bring out a clarity on actionable insights, and what is ICMR doing to ensure that India can make an Impact like China.	We thank the reviewer for his valuable contributions to help us improve our article. We hope the concerns have been allayed by the revisions we have made to the article.

Dear Reviewers,

We thank you for your valuable comments. The editions have improved our manuscript immeasurably. Please find below a point-by-point response to the second round of review in a tabular format. We have tried to stay true to our vision for the manuscript through the revisions made. Most of the suggestions have already been incorporated in the manuscript. Wherever we have been unable to incorporate the suggestions in the manuscript, we have elaborated on why it was difficult to do so. We are now submitting these responses for your kind review.

Regards,

Dr Aruvi Poomali

Corresponding author

Reviewer #1 comments	
Round 2 comments	Response
“While this may be a universal phenomenon, it worth noting that India...”	Has been changed to “While this may be a universal phenomenon, it is worth noting that India...”

Reviewer #2 comments			
Round 1 comments	1 st Response	Round 2 comments	2 nd Response
The number of Phase 1 clinical trials conducted in countries such as Japan, UK, Germany	Please find our response below: While the number of Phase 1 clinical trials conducted in	The Authors have not provided a suitable response for the following comment.	The submitted manuscript “National Assets for Early Phase Clinical Trials Capacity in India” was envisioned to

and Australia is very similar to India.  • Are regulatory challenges, lack of infrastructure and lack of skilled personnel the only obstacles that need to be targeted. • If so then why do these countries lag behind? • We must also think of global level challenges that are not generic to developing countries. • What will be done by ICMR to encounter them? • What can India do differently than Japan, UK, Germany and Australia? 	countries such as Japan, UK, Germany and Australia appear similar to India, we believe this could be due to a variety of reasons. A study has previously shown possible misclassifications and inaccuracies in one such registry and reports that only 220 First-in-human phase 1 clinical trials were conducted in India between 2008-2022. (reference #7) Notwithstanding these inaccuracies, the countries would still remain incomparable without an in-depth analysis the proportion of funding that goes into pharmaceutical R&D and the percentage of GDP that is earmarked for drug development. At the European Federation for Exploratory Medicines Development Lyon Conference 2019, experts identified the changing landscape of early clinical development and the challenges created by		describe the ICMR Network for Phase 1 Clinical Trials, its vision, governance, and operational aspects.  • In our rebuttal we clarify that even though the number of Phase 1 clinical trials conducted in these countries may look similar, the reality may be different. (reference #7) • In addition, the challenges to conducting clinical trials have been identified by other reviewers (References in 1st response) and is also a research question that has been taken up by our study team and is beyond the scope of this manuscript. • We believe that to do justice to the question on why these countries lag behind, we will need to take it up as a research question. It would require an in-depth analysis of the clinical trial and drug development landscape in each of the four countries, the identification of challenges to clinical trial in each country and analysis of why they
---	--	--	---

	transformative technologies that are increasingly a part of trial designs. Newer agents such as biologicals also present challenges in early phase drug development. https://www.frontiersin.org/journals/pharmacology/articles/10.3389/fphar.2019.01377/full) However, the CIOMS 2021 consensus on “Clinical research in resource-limited settings” identify a lack of conducive environment, challenges in implementing international standards and trust building as the major hurdles to good quality clinical research. https://cioms.ch/publications/product/clinical-research-in-low-resource-settings/) Further, a work of Burt et al. (referenced in our article) has extensively researched the challenges to the clinical research environment in India. The domains they have identified include Regulatory		have not been able to identify solutions and pull ahead. This might be beyond the scope of the current manuscript.  • On the same note, exploring the global level challenges to drug development is a vast subject as discussed at the 2019 Lyon conference referenced in 1st response and is beyond the scope of this manuscript. • ICMR is currently in the process of identifying the granular details of the challenges in conducting early phase clinical trials. • India has already started working differently by setting up the Phase I Clinical Trial Network and the INTENT network for late phase clinical trials. Further comment can be made only after experience into the drug development landscape of these countries as described above. The learnings from the experience of implementing trials will be published in the coming years.
--	--	--	---

	(government, ethics committees, monitors, auditors), Professional (industry, academia, health care: clinicians, investigators, research staff), and Public (patient advocacy groups, NGOs, the media).(Reference #24) Thus we can see that the challenges faced by the high income countries in early phase clinical trials might be economic as well as scientific and we believe that the ICMR Network for Phase I clinical trials can address this through capacity building and support for the development of molecules of national health priority.		
What will be done to ensure that the Phase I trials conducted in India meet the Global level regulatory requirements?	Please find our response below: The ICMR Central Coordinating Unit has so far conducted two major workshops for the ICMR Network for Phase 1 Clinical Trials. The first workshop was on quality systems and SOPs. The	The authors have submitted a rebuttal letter addressing the provided comments. However, for certain comments, the authors have only provided responses in the letter, without incorporating the	 • The capacity building taken up by the network has been incorporated in the manuscript as shown in the italicized extract in the 1st Response. • ICMR is establishing capacity to conduct Clinical Trials of international standards by setting up the ICMR Network for Phase 1

	sites have since then harmonized the SOPs across all centres. The second workshop titled “The ICMR-PGIMER Workshop of Phase 1 Clinical Trials on Regulatory Aspects” was held with experts from two regulatory agencies (CDSCO and FDA) and industry leaders. The roadmap for future workshops and periodic in-house training for all sites has also been set up and has been mentioned in the text: “The infrastructure development phase has been completed, SOPs have been developed to harmonize practices, and a road map for workshops and periodic in-house training for all sites has been set up”.	corresponding content into the manuscript itself.	Clinical Trials. The network is expected to cater to clinical trials for regulatory submission across the world. The regulatory requirement of the Central Licencing Authority of each country is different and the Network is expected to be flexible enough to meet these requirements.
“India's decision to waive local clinical trials for drugs approved in countries like the U.S., U.K., Japan, and the European Union represents a pivotal regulatory	Please find our response below: “The ICMR Network of Phase 1 Clinical Trials is designed to create a robust infrastructure that	The authors have submitted a rebuttal letter addressing the provided comments. However, for certain comments, the authors have	 • The submitted manuscript “National Assets for Early Phase Clinical Trials Capacity in India” was envisioned to describe the ICMR Network for Phase

shift. While the move aims to expedite access to essential medications and enhance market availability, it raises significant concerns about patient safety, research and development (R&D), and broader healthcare implications.” – How will this impact the ICMR Network of Phase 1 clinical Trials?	supports the rigorous demands of early phase clinical trials and aims to foster a conducive environment for these trials in India, and thereby contribute to global drug discovery and development. Utilization of this capacity built by ICMR is expected to derisk the failure inherent in early phase product development.” The Network is expected to aid in infrastructure development, manpower training, standardization of processes, collaborations and partnerships, support for innovators and cost and risk sharing. According to the authors, India’s decision to waive local clinical trials for drugs approved in countries for five categories of drugs (namely orphan drugs for rare diseases, gene and cellular therapies, pandemic related drugs, defense-specific treatments and drugs with significant therapeutic	only provided responses in the letter, without incorporating the corresponding content into the manuscript itself.	1 Clinical Trials, its vision, governance, and operational aspects.  • As the reviewer has mentioned, the pivotal regulatory shift expedites access to medication and has broad healthcare implications. • We believe that the current manuscript would not be the ideal place for a discussion on implications of regulatory changes.
---	---	---	---

	advancements) is not expected to impact the activities of Phase 1 clinical trials. While the decision will improve access to certain drugs under certain condition, the Network, moving on a different trajectory will help build indigenous capacity for drug development.		
What is the long-term global implication of this initiative?	Please find our response below: Once established, the ICMR Network for Phase 1 clinical trials will not only build capacity for development of molecules of national priority but also be a beacon for drug development for LMICs, especially in the South-East Asian region. In their paper, Pant et al. elaborate on how the emerging Indian model - which includes the ICMR Network for Phase 1 Clinical Trials - will help LMICs in south east Asia breakaway from their dependency on high income	The authors have submitted a rebuttal letter addressing the provided comments. However, for certain comments, the authors have only provided responses in the letter, without incorporating the corresponding content into the manuscript itself.	The manuscript extract with the vision of this initiative is given below. “The ICMR Network of Phase 1 Clinical Trials is designed to create a robust infrastructure that supports the rigorous demands of early phase clinical trials and aims to foster a conducive environment for these trials in India, and thereby contribute to global drug discovery and development. Utilization of this capacity built by ICMR is expected to derisk the failure inherent in early phase product development.”

	countries for solutions to their national health problems.(Reference #16) In addition, the network is also expected to make India a go – to destination for early phase clinical trials from across the world.		While we envision the project to contribute to global drug development and have responded previously on how the project might become a beacon for LMICs as has already been described in another paper (Referenced in 1st Response), to further elaborate on the long term global implications in the manuscript would be conjecture at this point. We believe that to satisfactorily answer the question, we should allow the project to mature and then analyze the long term implications based on data reflecting therapy areas, origin of collaborators, etc.
Can Phase 1 clinical trials of Global Pharmaceutical MNC's be outsourced to India?	Please find our response below: The ICMR Network for Phase 1 Clinical Trials is expected to make India a go – to destination for early phase clinical trials from across the world. While the	The authors have submitted a rebuttal letter addressing the provided comments. However, for certain comments, the authors have only provided responses in the letter, without incorporating the	This has been mentioned in the manuscript as shown below. Collaborations can be with pharmaceutical companies, agnostic of origin. “4. Collaborations and Partnerships: The initiative also emphasizes the importance

	centres of the Network are expected to prioritize the molecules of national health priority identified and funded by ICMR, the centres are open to all early phase drug development agnostic of their origin.	corresponding content into the manuscript itself.	of collaborations with research institutions and pharmaceutical companies, agnostic of their origin. By fostering these partnerships with innovators, the ICMR aims to enhance knowledge exchange and integrate global advancements into India's research framework.”
Majority of the support only appears to be offered to “Academia and start-ups with limited experience”. What kind of support will be offered by the ICMR to bigger pharmaceutical companies?	Please find our response below: While the ICMR Network for Phase 1 Clinical Trials is expected to support innovators with limited experience, the centres are open to all early phase drug development agnostic of their origin. The manuscript has been edited to reflect this. “4. Collaborations and Partnerships: The initiative also emphasizes the importance of collaborations with research institutions and pharmaceutical companies, agnostic of their origin. By fostering these partnerships with innovators, the	The authors have submitted a rebuttal letter addressing the provided comments. However, for certain comments, the authors have only provided responses in the letter, without incorporating the corresponding content into the manuscript itself.	The extract from the manuscript given below shows that collaborations and partnerships other than academia and start-ups are also encouraged. The authors have addressed the potential collaboration with suitable partners including big pharmaceutical companies. However, since many of these companies have their own mechanisms to take forward their leads, the priority for the ICMR Phase I clinical trial network is to collaborate with academic and start-ups with limited experience in areas of national health priority. The revised text is provided in Response #1.

	ICMR aims to enhance knowledge exchange and integrate global advancements into India's research framework. 5.Support for innovators: The initiative is expected to extend regulatory and operational support to innovators, especially to academia and start-ups with limited experience in early phase drug development.”		
--	--	--	--

Dear Reviewers,

We thank you for your valuable comments. Please find below a point-by-point response to the third round of review in a tabular format for your kind review.

Regards,

Dr Aruvi Poomali

Corresponding author

Round 3 comments – Reviewer 2	Response
1. The authors have declined to discuss regarding the global level challenges which developed countries who have the same number of Phase 1 Trials as India face – “stating that it is beyond the scope of the manuscript”. Such a rebuttal is not acceptable in an International Journal that caters to the Global Community of HCP’s	The authors would like to reiterate that while the question the reviewer has shared is important, the submitted manuscript “National Assets for Early Phase Clinical Trials Capacity in India” might not be the appropriate place to bring out the discussion on the same. However, our ongoing review of early phase clinical trial regulations in various geographies might serve as an appropriate place to discuss challenges in establishing Phase 1 capacity in the country. We assure the reviewers that the comment will be addressed in detail in the upcoming paper. Also, our manuscript is currently longer than the typical length of a “Comment” on Nature Communications Medicine (typically 1-4 pages, up to 1,500 words) and request your kind consideration.
2. “The network is expected to cater to clinical trials for regulatory submission across the world. The regulatory requirement of the Central Licencing Authority of each country is different and the	We thank the reviewer for this suggestion. The same has been now included in the manuscript (available in Table 1).

Network is expected to be flexible enough to meet these requirements.” – This is just mentioned in the rebuttal letter and not in the manuscript.	
--	--